

# Exploring the perception and readiness of Pharmacists towards telepharmacy implementation; a cross sectional analysis

Khayal Muhammad[1,2], Mohamed A. Baraka[3,4], Syed Sikandar Shah[5], Muhammad Hammad Butt[6], Haytham Wali[7], Muhammad Saqlain[8], Tauqeer Hussain Mallhi[9], Khezar Hayat[10], Khairi Mustafa Fahelelbom[11], Royes Joseph[12] and Yusra Habib Khan[9]

[1] Department of Clinical Pharmacy, Faculty of Pharmacy, Near East University, Northern Cyprus
[2] Faculty of Pharmaceutical Sciences, Abasyn University, Peshawar, KPK, Pakistan
[3] Clinical Pharmacy Program, College of Pharmacy, Al Ain University, Al-Ain, Abu Dhabi, United Arab Emirates
[4] Clinical Pharmacy Department, College of Pharmacy, Al-Azhar University, Nasr city, Cairo, Egypt
[5] Department of Clinical Pharmacy, Faculty of Pharmacy, European University of Lefke, Lefke, Cyprus
[6] Faculty of Pharmacy, University of Central Punjab, Lahore, Punjab, Pakistan
[7] Department of Pharmacy Practice, College of Clinical Pharmacy, King Faisal University, Al-Ahsa, Saudi Arabia
[8] Department of Pharmacy, Quaid-i-Azam University, Islamabad, Pakistan
[9] Department of Clinical Pharmacy, College of Pharmacy, Jouf University, Sakaka, Al-Jouf Province, Kingdom of Saudi Arabia
[10] Department of Pharmacy Administration and Clinical Pharmacy, School of Pharmacy, Xi'an Jiaotong University, Xi'an, China
[11] Pharmaceutical Sciences Program, College of Pharmacy, Al-Ain University, Al Ain, Abu Dhabi, United Arab Emirates
[12] Department of Pharmacy Practice, College of Clinical Pharmacy, Imam Abdulrahman Bin Faisal University, Dammam, Saudi Arabia

Corresponding authors
Tauqeer Hussain Mallhi,
tauqeer.hussain.mallhi@hotmail.com
thhussain@ju.edu.sa
Muhammad Hammad Butt,
hmdbut@ucp.edu.pk

## ABSTRACT

**Background:** Amid the turbulent nature of the COVID-19 pandemic, telepharmacy has shifted the paradigm of patient care by leveraging digital medicine. Government mandated lockdowns and norms of social distancing have further underscored the need for telepharmacy. Many developed and developing countries implemented such initiatives where pharmacists have provided tele-pharmacy services *via* telecommunications. However, the implementation and utilization of tele-pharmacy services are quite negligible in resource limited settings due to financial and administrative constraints. This study was aimed to ascertain the perception and readiness of pharmacists working in various sectors of a resource limiting country.

**Methodology:** A cross sectional study was carried out in all provinces of Pakistan to explore the perceptions of pharmacists towards telepharmacy implementation through a 35-items study instrument. The collected data was analyzed descriptively and scored accordingly. The chi-square test was used for inferential analysis on pharmacist's perception regarding implementation of tele-pharmacy with their demographics.

**Results:** Of 380 pharmacists, the mean age is 27.67 ± 3.67 years with a preponderance of male pharmacists ($n = 238$, 62.6%). The pharmacists ($n = 321$, 84.5%) perceived that telepharmacy implementation improves patient's quality of life and decreases

patients' visits ($n$ = 291, 76.6%). Overall, pharmacists ($n$ = 227, 59.7%) had negative perception towards benefits of telepharmacy implementation, but pharmacists had positive perception towards eligibility ($n$ = 258, 67.9%), regulatory issues ($n$ = 271, 71.3%) and telepharmacy during pandemic and beyond ($n$ = 312, 82.1%).

In chi-square testing gender ($p$ = 0.03) and age ($p$ = 0.03) had a significant association with perception regarding regulatory issues. Among perception regarding telepharmacy during COVID-19 pandemic and beyond age had a significant association ($p$ = 0.03). Among perception regarding eligibility job location of pharmacists had significant association ($p$ = 0.04).

**Conclusion:** The majority of pharmacists had a positive perception regarding the eligibility of patients and regulatory issues/legal framework regarding the implementation of tele-pharmacy, as well as its use during the COVID-19 pandemic and beyond. The implementation of tele-pharmacy can play a major role in providing timely and better patient care to remote patient areas and may help in the prevention and treatment of different infectious diseases.

# INTRODUCTION

The coronavirus disease 2019 (COVID-19) was spread throughout the world and disrupted global public health, causing irreversible loss and increasing humanistic effect (*Misbah et al., 2020*). As of March 4, 2022, the world witnessed more than 440,807,756 confirmed COVID-19 cases and a total death of 5,978,096 (*World Health Organization, 2022*). Moreover, developed and developing countries have had to implement strict measures such as early diagnoses, health quarantine, control measures, social distancing, lockdown, use of hand sanitizers and face masks to retain the efficacy of their interrupted health care systems (*Butt et al., 2021*; *Mohamed Ibrahim et al., 2020*).

The practice of providing pharmaceutical care to patients during the pandemic has traditionally been limited to simple physical examination and treatment. Doctors were bound to provide health care services to the patients in specific areas such as private clinics and hospitals. Recently, with such a global issue, the traditional way of providing care services to patients has been modified, and a new method of care services called telemedicine has been developed, called telemedicine. Telemedicine is the modern way of providing health care services to patients and the general community *via* different telecommunication tools like telephones, smartphones, tablets, wireless devices, and laptops (*Dorsey & Topol, 2016*). Previously, just 250,000 patients used telemedicine services, but with time and improved awareness, it has reached up to 3.2 million patients (*Vecchione, 2016*).

Telemedicine services include telehealth and remote patient monitoring. Remote patient monitoring *via* web applications has been studied to improve patient-reported quality of life, medication adherence, and decrease health care costs (*George & Cross, 2020*). Telepharmacy falls under the umbrella of telemedicine and is the provision of

pharmaceutical care to patients using technologies and telecommunications (*Le, Toscani & Colaizzi, 2020*). Telepharmacy allows the pharmacist to provide patient care services in a remote area by taking a history, reviewing patients' files, and assessing medications utilized by the patients. This practice also helps the technician to dispense patient prescriptions accurately under the remote supervision of the pharmacist. It provides opportunities for improving health outcomes for patients and the quality of health care systems in general. The use of telepharmacy includes patient counseling, mail order of medicines, drug management, medication therapy management, supervision of technician dispensing, central processing, and automated dispensing systems, with pharmacist tele-counseling (*Vecchione, 2016*).

In the United States of America (USA), community pharmacies in rural areas adopt telepharmacy to improve access to pharmaceutical care services for their patients (*Kimber & Peterson, 2006*). Telepharmacy can help in the combating and managing different diseases, such as asthma, COVID-19, and Dengue in different settings. The implementation of telepharmacy in hospitals can also limit the burden of disease on patients, as well as limit potential adverse drug events (*Brown et al., 2017*; *Schneider, 2013*). A study reported that pharmacist interventions could increase up to 42% by implementing telepharmacy services. The extended use of telepharmacy services increases patient access to pharmacy services and consultation of patients as well (*Ibrahim et al., 2020*).

During the COVID-19 pandemic, numerous countries, including the United Kingdom, Canada, Australia, and the USA, allowed pharmacists to provide virtual consultation services through different sources, such as websites, home delivery of prescription drugs, and permission to compound antiseptic solutions (*Gross & MacDougall, 2020*; *Merks et al., 2020*). The United Arab Emirates allowed health care professionals to regulate telepharmacy services to help in combating the COVID-19 pandemic (*Ibrahim et al., 2020*). These services include the distribution of medications, food supplements, herbal supplements, patient counselling, cosmetic products, labeling systems, and automatic packaging (*Dubai Health Authority, 2020*). Legalization of such services is important to help the community pharmacist to utilize telepharmacy in identifying symptoms of COVID-19 and seasonal flu among patients, providing appropriate testing facilities and basic information to lessen the viral spread among individuals (*Collins & Moles, 2019*; *Adunlin, Murphy & Manis, 2020*).

In China, 19 hospitals jointly developed "cloud-pharmacy care" services to help patients to discuss their medication related issues with pharmacists using the internet and texts (*Liu et al., 2020*; *Li et al., 2021*). Similarly, in China, a hospital also established "zero-contact pharmaceutical care" services during the COVID-19 pandemic (*Hua et al., 2020*).
In Saudi Arabia, an institute implemented e-tools such as webpage portals and WhatsApp business application to provide consultation services (*Asseri et al., 2020*). A study from Sri Lanka reported that the use of telemedicine services, audio consultations, and videoconferencing helped patients in the COVID-19 pandemic (*Kulatunga et al., 2020*). In the United States of America (USA), a hospital used Microsoft Teams for patients counseling and educating them regarding the usage of medications (*Elson et al., 2020*).

In the USA, the Department of Health & Human Services and state pharmacy boards temporarily updated the HIPAA rules and allowed telepharmacy services in conjunction with telemedicine by allowing videoconferencing (*Elson et al., 2020*; *Segal et al., 2020*). Verbal consent before the start of consultation is essential for telemedicine and telepharmacy services (*Margusino-Framiñán et al., 2020*; *Tortajada-Goitia et al., 2020*). Furthermore, social networks and online discussion boards could be useful tactics for telehealth services (*Crilly & Kayyali, 2020*). A study reported the use of hotline numbers for online consultations and prescription orders in New Zealand, the use of fax, email, and mail for prescription purposes, mobile applications for home delivery of medications in Colombia and China, and in Pakistan, telepharmacy services were provided for basic consultation and triaging (*Bukhari, Rasheed & Nayyer, 2020*).

In literature, numerous studies were conducted in urban and rural areas of Pakistan related to telemedicine and telepharmacy implementation. A study checked the effectiveness of telepharmacy training and found that knowledge and training of telecommunication are necessary to provide telehealth services. Telepharmacy interventions can be provided using short message services (SMS) from mobile phone, video conferencing, mobile health (mHealth)-based applications, and web-based services (*Gul et al., 2008*). The telepharmacy services were provided to patients suffered with chronic diseases such as hypertension stroke and diabetes. The patients were provided care by using mHealth services, call remainders, SMS at home (*Siddiqui et al., 2015*; *Iftikhar et al., 2019*; *Kamal et al., 2020*). The telehealth services were also effectively used for vaccine campaigns and help regulatory authorities to make sure public is fully vaccinated (*Kazi et al., 2017*).

Telepharmacy services raised several concerns about patient privacy and the nature of patient data, as an electronic record can save individuals' details easily than documentation on paper. Therefore, ensuring the privacy of patient data must be prioritized and protection must be maintained by health care professionals, including pharmacists (*Crico et al., 2018*). The telemedicine studies were conducted among doctors (*Ashfaq et al., 2020*), and university students (*Tariq, Khan & Basharat, 2020*) before the COVID-19 pandemic but no study was conducted among pharmacists towards telepharmacy implementation during the COVID-19 pandemic. Considering this we designed the first nationwide study that aimed to explore the perception of pharmacists towards telepharmacy implementation during the COVID-19 pandemic in Pakistan.

## MATERIALS AND METHODS

### Ethical approval

The study was officially approved by the ethics committee of Riphah International University, Islamabad (Reference Number: EC-RIP/271/0203). The consent to participate in the study was taken in the Google form. The participant who provided consent to participate can fill the study questionnaire. A preface on the study was available before filling the questionnaire that describes the nature and purpose of the study including consent part that ensures anonymity and voluntary participation of respondents.

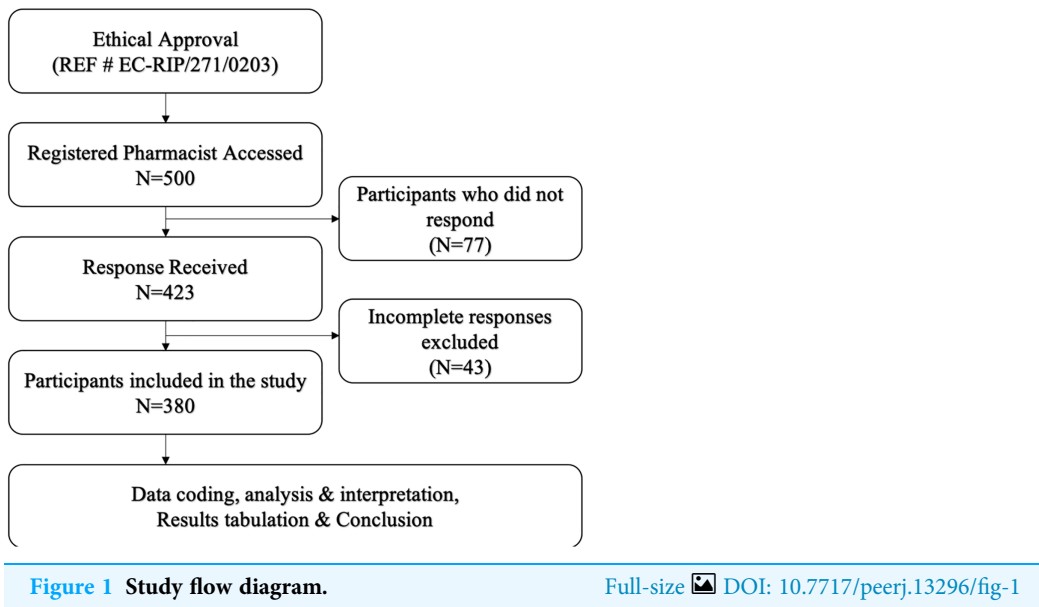

**Figure 1  Study flow diagram.**              

## Study design

A cross-sectional study was conducted from 1st of March, 2021 to 1st of May, 2021 using an online survey due to the government restrictions in accessing healthcare facilities during the pandemic. In the current pandemic, it was impossible to conduct a community-based sampling survey, so we decided to collect data from pharmacists through social media platforms. The study flow diagram is presented in Fig. 1.

## Study population and sampling

The registered pharmacists from the pharmacy council of all provinces working in different settings (hospital pharmacy, clinical setup, community pharmacy, etc.) were targeted to participate in this survey. The questionnaire was sent to pharmacists individually or through social media channels including Facebook, WhatsApp, Twitter, and LinkedIn platforms. There are more than 60 groups, pages or accounts related to the Pakistani pharmacists on Facebook and Twitter. Provincial social groups on Facebook were also considered in data collection. A link of survey along with brief description of the study was posted in all groups with admin approval. Considering the snowball sampling technique, the survey link was also shared with officials of national and provincial pharmacist societies so they could forward to pharmacists at city-level. A regular monitoring of responses was conducted and efforts were increased for provinces where response rate was low.

## Inclusion and exclusion criteria

All the registered pharmacists working in Pakistan and willing to voluntarily participate in the study were considered eligible for this survey. However, pharmacists working abroad or foreign license holders, pharmacy students who were not graduated yet, unemployed pharmacists, and those who don't want to participate were excluded from this study.

## Sample size

The sample size was calculated using an online sample size calculator named Raosoft sample size calculator (*Raosoft, 2004*), which comes out to be 380 with a population size of 34,000 licensed pharmacists in Pakistan. The response percentage to the survey was presumed to be 50%, the margin of error 5%, and 95% confidence level. Response acceptance was closed when the required sample size was achieved.

## Study questionnaire

A self-administered questionnaire-based study instrument was developed after an in-depth review of the literature on COVID-19. Initial version of the study tool was shared with five subject matter experts from pharmacy profession. All the suggestions were made in the questionnaire and revised draft was again shared with the experts. Subsequently, an opinion of experts against essentiality or usefulness of each item in the tool was requested and considered before finalizing the study instrument. Following a content and face validity, a pilot study was conducted on 30 pharmacists. The reliability coefficient was checked using SPSS version 21, and Cronbach alpha values come out to be 0.745. The study instrument consisted of five sections; demographics, telepharmacy implementation for patients, patients' eligibility for telepharmacy, regulatory issues, and legal framework for telepharmacy during COVID-19 pandemic and beyond.
The demographic section included job location, gender, age in years, and province of residence. The second section that was concerned with telepharmacy implementation for patients contained 11 questions (1-Improve the patients' quality of life, 2-Decrease the patient's visits to hospitals, 3-Decrease the rate of morbidity and mortality, 4-Improve patients' medication adherence, *etc.*). The third section on the eligibility of patients for telepharmacy consisted of three questions (1-Patients with infectious diseases, 2-Patients with chronic diseases, 3-Patients using multiple medications). The fourth section of regulatory issues and legal framework for telepharmacy contained 14 questions (1-Telepharmacy services should only be provided by a clinical pharmacist, 2-Requires special tools and space in the pharmacy, 3-Requires special training for pharmacists, etc.) and the final (fifth) section of telepharmacy during the COVID-19 pandemic and beyond comprises three questions (1-Telepharmacy implementation is a good approach, 2-Patients will be interested in receiving telepharmacy services after the pandemic, 3-I am familiar with the requirements of telepharmacy implementation). All responses to questions Section 2, 3, 4, 5 were recorded on a five-point "Likert scale" as strongly disagree = 1, disagree = 2, neutral = 3, agree = 4, strongly agree = 5. The complete questionnaire was presented in supplemental files.

## Statistical analysis

Data were collected as previously described in *Alotaibi et al. (2021)*. First, the collected data were added in the Microsoft excel sheet and coded for statistical analysis in SPSS version 22.0. The data of categorical variables were presented as number (*N*) and percentage (%), while continuous variables were presented as mean (M) and standard deviation (S.D). The chi-square test was used to compare the categorical variables from
**Table 1 Demographic characteristics.**

| Variables | | Frequencies | Percentages |
|---|---|---|---|
| Gender | Male | 238 | 62.6 |
| | Female | 142 | 37.4 |
| Age | 20–29 years | 283 | 74.5 |
| | 30–39 years | 90 | 23.7 |
| | 40 years & above | 7 | 1.8 |
| Job location | Community pharmacy | 121 | 31.8 |
| | Hospital pharmacy | 81 | 21.3 |
| | Other | 178 | 46.8 |
| Province | Baluchistan | 48 | 12.6 |
| | GB | 19 | 5 |
| | KPK | 135 | 35.5 |
| | Punjab | 111 | 29.2 |
| | Sindh | 67 | 17.6 |

demographic section with perception and attitude variables asked from the pharmacists towards telepharmacy implementation. The $p$-value of less than 0.05 in the chi-square test was considered as significance.

# RESULTS

A total of 380 pharmacists participated in the study: 62.6% ($n$ = 238) were males, and 31.8% ($n$ = 121) were from community pharmacies. The majority (64.7%) of them were from KPK or Punjab province. The mean age of the participants was (27.67 ± 3.67). Detailed demographic characteristics are available in Table 1.

Responses were recorded about the perceptions of different sections of telepharmacy implementation for all the questions. Approximately 84.5% of pharmacists have agreed that the implementation of telepharmacy could improve the patient's quality of life, and 76.6% agreed that its implementation could decrease patient visits to hospitals, private clinics, or pharmacies. Similarly, 80% of the pharmacists agreed that telepharmacy implementation could improve patients' medication adherence, patient disease therapy management and can help in pharmaceutical care provision by preventing disease transmission. Almost 82.3% of the pharmacists agreed that telepharmacy implementation could increase patients' appreciation of the pharmacist's role. The pharmacists who agreed that the patients using multiple medications and patients with infectious diseases are the most eligible patients for telepharmacy implementation were 68.8% and 43.2% respectively. 81.6% of pharmacists agreed that a legal collaboration agreement between pharmacists, physicians, and other healthcare providers is required, while 82.2% agreed that the availability of adequate drug information services and resources is required for regulatory issues and legal framework for telepharmacy. Moreover, 81.8% of the pharmacists agreed that telepharmacy implementation is a good approach during the COVID-19 pandemic and beyond. The detailed results are presented in Table 2.

**Table 2 Summary of pharmacists' perception about the telepharmacy implementation.**

| Questions | | Strongly disagree | | Disagree | | Neutral | | Agree | | Strongly dgree | |
|---|---|---|---|---|---|---|---|---|---|---|---|
| | | N | % | N | % | N | % | N | % | N | % |
| **Telepharmacy benefits** | | | | | | | | | | | |
| BQ1 | Improve the patients' quality of life | 7 | 1.8 | 11 | 2.9 | 41 | 10.8 | 230 | 60.5 | 91 | 23.9 |
| BQ2 | Decrease the patient's visits to hospitals, private clinics, or pharmacies | 7 | 1.8 | 22 | 5.8 | 60 | 15.8 | 223 | 58.7 | 68 | 17.9 |
| BQ3 | Decrease the rate of morbidity and mortality in patients | 4 | 1.1 | 30 | 7.9 | 104 | 27.4 | 194 | 51.1 | 48 | 12.6 |
| BQ4 | Improve patients' medication adherence | 6 | 1.6 | 23 | 6.1 | 52 | 13.7 | 240 | 63.2 | 59 | 15.5 |
| BQ5 | Improve patient disease therapy management | 12 | 3.2 | 15 | 3.9 | 48 | 12.6 | 240 | 63.2 | 65 | 17.1 |
| BQ6 | Help in the identification, resolution or prevention of drug-related problems | 6 | 1.6 | 31 | 8.2 | 58 | 15.3 | 220 | 57.9 | 65 | 17.1 |
| BQ7 | Help in pharmaceutical care provision by preventing disease transmission | 5 | 1.3 | 25 | 6.6 | 44 | 11.6 | 240 | 63.2 | 66 | 17.4 |
| BQ8 | Increase the level of job satisfaction among pharmacists | 9 | 2.4 | 23 | 6.1 | 69 | 18.2 | 203 | 53.4 | 76 | 20.0 |
| BQ9 | Increase patients' appreciation to the pharmacist's role | 6 | 1.6 | 15 | 3.9 | 46 | 12.1 | 214 | 56.3 | 99 | 26.1 |
| BQ10 | Will increase job opportunities for pharmacists | 13 | 3.4 | 14 | 3.7 | 61 | 16.1 | 206 | 54.2 | 86 | 22.6 |
| BQ11 | Reduce the burnout rate of health care providers, especially protection in a pandemic | 8 | 2.1 | 12 | 3.2 | 70 | 18.4 | 218 | 57.4 | 72 | 18.9 |
| **Eligibility of patients** | | | | | | | | | | | |
| EQ1 | Patients with infectious diseases | 19 | 5.0 | 126 | 33.2 | 71 | 18.7 | 137 | 36.1 | 27 | 7.1 |
| EQ2 | Patients with chronic diseases | 19 | 5.0 | 118 | 31.1 | 88 | 23.2 | 126 | 33.2 | 29 | 7.6 |
| EQ3 | Patients using multiple medications | 10 | 2.6 | 42 | 11.1 | 97 | 25.5 | 183 | 48.2 | 48 | 12.6 |
| **Regulatory issues and legal frameworks for telepharmacy** | | | | | | | | | | | |
| RQ1 | Legal collaboration agreement between pharmacists, physicians, and other healthcare providers | 7 | 1.8 | 13 | 3.4 | 50 | 13.2 | 202 | 53.2 | 108 | 28.4 |
| RQ2 | Telepharmacy services should only be provided by a clinical pharmacist | 13 | 3.4 | 58 | 15.3 | 77 | 20.3 | 165 | 43.4 | 67 | 17.6 |
| RQ3 | Pharmacists should focus on drug-dispensing services only and leave providing telehealth services for physicians | 60 | 15.8 | 95 | 25.0 | 51 | 13.4 | 131 | 34.5 | 43 | 11.3 |
| RQ4 | Facilitating the access of pharmacists to the patients' medical records | 5 | 1.3 | 9 | 2.4 | 54 | 14.2 | 230 | 60.5 | 82 | 21.6 |
| RQ5 | Requires special tools and space in the pharmacy | 5 | 1.3 | 25 | 6.6 | 66 | 17.4 | 220 | 57.9 | 64 | 16.8 |
| RQ6 | Time limitation could be a barrier | 9 | 2.4 | 42 | 11.1 | 92 | 24.2 | 192 | 50.5 | 45 | 11.8 |
| RQ7 | Unnecessary extra load for the pharmacists | 32 | 8.4 | 149 | 39.2 | 59 | 15.5 | 109 | 28.7 | 31 | 8.2 |
| RQ8 | More comfortable seeing the patient face-to-face than through telepharmacy | 10 | 2.6 | 33 | 8.7 | 85 | 22.4 | 189 | 49.7 | 63 | 16.6 |
| RQ9 | Automate away the social and empathic aspects of care, decreasing its therapeutic value | 14 | 3.7 | 63 | 16.6 | 119 | 31.3 | 149 | 39.2 | 35 | 9.2 |
| RQ10 | Requires increase in the number of pharmacists | 13 | 3.4 | 12 | 3.2 | 60 | 15.8 | 240 | 63.2 | 55 | 14.5 |
| RQ11 | Requires special training for pharmacists | 6 | 1.6 | 18 | 4.7 | 58 | 15.3 | 210 | 55.3 | 88 | 23.2 |
| RQ12 | Requires the availability of adequate drug information services and resources | 6 | 1.6 | 6 | 1.6 | 48 | 12.6 | 230 | 60.5 | 90 | 23.7 |
| RQ13 | Require a repayment system for pharmacists | 6 | 1.6 | 15 | 3.9 | 48 | 12.6 | 238 | 62.6 | 73 | 19.2 |
| RQ14 | Prepared for telepharmacy implementation | 7 | 1.8 | 19 | 5.0 | 86 | 22.6 | 192 | 50.5 | 76 | 20.0 |
| **Telepharmacy in COVID-19 pandemic and beyond** | | | | | | | | | | | |
| PBQ1 | Telepharmacy implementation is a good approach during the COVID-19 pandemic and beyond | 11 | 2.9 | 9 | 2.4 | 49 | 12.9 | 193 | 50.8 | 118 | 31.1 |
| PBQ2 | Patients will be interested in receiving telepharmacy services after the pandemic | 12 | 3.2 | 20 | 5.3 | 72 | 18.9 | 212 | 55.8 | 64 | 16.8 |
| PBQ3 | I am familiar with the requirements of telepharmacy implementation | 14 | 3.7 | 32 | 8.4 | 99 | 26.1 | 191 | 50.3 | 44 | 11.6 |

**Note:**
B, Benefit; E, Eligibility; R, Regulatory; PB, Pandemic and Beyond.

**Table 3 Perceptions regarding the telepharmacy implementation.**

| Variable | Categories | Frequency | Percentages |
|---|---|---|---|
| Perception regarding benefits | Negative (11–38) | 227 | 59.7 |
| | Positive (39–55) | 153 | 40.3 |
| Perception regarding eligibility | Negative (5–10) | 122 | 32.1 |
| | Positive (11–15) | 258 | 67.9 |
| Perception regarding regulatory issues and legal frameworks | Negative (14–48) | 109 | 28.7 |
| | Positive (49–70) | 271 | 71.3 |
| Perception regarding telepharmacy in COVID-19 pandemic and beyond | Negative (5–10) | 68 | 17.9 |
| | Positive (11–15) | 312 | 82.1 |

The overall perception regarding different items of telepharmacy implementation is presented in Table 3. Within the total of participants, 71.3% ($n$ = 271) had positive perceptions regarding regulatory issues and legal frameworks for telepharmacy, while 82.1% ($n$ = 312) of pharmacists had a positive perception about the implementation of telepharmacy in the COVID-19 pandemic and beyond.

The pharmacist's perception towards the regulatory issues of telepharmacy implementation was significantly associated with gender as most male respondents, ($n$ = 171; 71.8%), ($p$ = 0.03), had a positive attitude compared to female (61.3%). The proportion of respondents who had a positive attitude towards regulatory issues was also significantly associated with age, as age was significantly related to the perceptions of the respondents about the implementation of telepharmacy in COVID-19 and beyond ($p$ = 0.03) and with perception about regulatory issues ($p$ = 0.03). It was evident that perception regarding eligibility was significantly associated to the job location ($p$ = 0.04) as a higher proportion of pharmacists working at other places showed positive perception compared to hospital and community pharmacies. The detailed information about the difference in perception regarding telepharmacy implantation by sample characteristics were presented in Table 4.

## DISCUSSION

This is the first nationwide study conducted to ascertain the pharmacists' views about the benefits of telepharmacy and regulatory barriers in its implementation in Pakistan. Most of the respondents agreed that telepharmacy could improve clinical outcomes in patients and limit their hospitalization. Literature reported that implementation of telepharmacy and telehealth added value in the selection of medication, dispensing, patient counseling, monitoring, and the provision of clinical services among patients (*Alexander et al., 2017*). A recent qualitative review on the current status of telemedicine in Pakistan underscored the need for training on telehealth services among healthcare professionals, as these services were found be to average in the healthcare system of Pakistan. This review also indicated the motivation of healthcare professionals and students in Pakistan towards the utilization of telemedicine services and electronic applications, particularly for rural population, vaccination campaigns and for those having

**Table 4 Difference in perception regarding telepharmacy implantation by sample characteristics.**

| Variables | Perception regarding telepharmacy benefits | | | | Perception regarding eligibility | | | | Perception regarding regulatory issues | | | | Perception regarding telepharmacy in COVID-19 pandemic and beyond | | | |
|---|---|---|---|---|---|---|---|---|---|---|---|---|---|---|---|---|
| | Negative N (%) | Positive N (%) | χ2 | p | Negative N (%) | Positive N (%) | χ2 | p | Negative N (%) | Positive N (%) | χ2 | p | Negative N (%) | Positive N (%) | χ2 | p |
| Gender | | | 0.98 | 0.321 | | | 3.12 | 0.07 | | | **4.56** | **0.03** | | | 2.90 | 0.08 |
| Female | 29 (20.4) | 113 (79.6) | | | 93 (65.5) | 49 (34.5) | | | 55 (38.7) | 87 (61.3) | | | 48 (33.8) | 94 (66.2) | | |
| Male | 39 (16.4) | 199 (83.6) | | | 134 (56.3) | 104 (43.7) | | | 67 (28.2) | 171 (71.8) | | | 61 (25.6) | 177 (74.4) | | |
| **Age** | | | 1.06 | 0.59 | | | 4.74 | 0.09 | | | **6.18** | **0.03** | | | **6.90** | **0.03** |
| 20–29 years | 54 (19.1) | 229 (80.9) | | | 178 (62.9) | 105 (37.1) | | | 100 (35.3) | 183 (64.7) | | | 91 (32.2) | 192 (67.8) | | |
| 30–39 years | 13 (14.4) | 77 (85.6) | | | 45 (50.0) | 45 (50.0) | | | 19 (21.1) | 71 (78.9) | | | 169 (17.8) | 74 (82.2) | | |
| 40 years & above | 1 (14.3) | 6 (85.7) | | | 4 (57.1) | 3 (42.9) | | | 3 (42.9) | 4 (57.1) | | | 2 (28.6) | 5 (71.4) | | |
| Job location | | | 0.94 | 0.62 | | | **6.02** | **0.04** | | | 1.08 | 0.58 | | | 1.83 | 0.41 |
| Community Pharmacy | 25 (20.7) | 96 (79.3) | | | 66 (54.5) | 55 (45.5) | | | 35 (28.9) | 86 (71.1) | | | 39 (32.2) | 82 (67.8) | | |
| Hospital Pharmacy | 13 (16.0) | 68 (84.0) | | | 43 (53.1) | 38 (46.9) | | | 29 (35.8) | 52 (64.2) | | | 19 (23.5) | 62 (76.5) | | |
| Others | 30 (16.9) | 148 (83.1) | | | 118 (66.3) | 60 (33.7) | | | 58 (32.6) | 120 (67.4) | | | 51 (28.7) | 127 (71.3) | | |
| Province | | | 4.72 | 0.31 | | | 4.19 | 0.38 | | | 1.19 | 0.88 | | | 4.14 | 0.39 |
| Baluchistan | 9 (18.8) | 39 (81.2) | | | 25 (52.1) | 23 (47.9) | | | 16 (33.3) | 32 (66.7) | | | 12 (25.0) | 36 (75.0) | | |
| GB | 5 (26.3) | 14 (73.7) | | | 14 (73.7) | 5 (26.3) | | | 5 (26.3) | 14 (73.7) | | | 6 (31.6) | 13 (68.4) | | |
| KPK | 17 (12.6) | 118 (87.4) | | | 76 (56.3) | 59 (43.7) | | | 40 (29.6) | 95 (70.4) | | | 32 (23.7) | 103 (76.3) | | |
| Punjab | 22 (19.8) | 89 (80.2) | | | 69 (62.2) | 42 (37.8) | | | 39 (35.1) | 72 (64.9) | | | 35 (31.5) | 76 (68.5) | | |
| Sindh | 15 (22.4) | 52 (77.6) | | | 43 (64.2) | 24 (35.8) | | | 22 (32.8) | 45 (67.2) | | | 24 (35.8) | 43 (64.2) | | |

**Note:**
Bold represent significant results with *p*-value <0.05.

chronic diseases (*Mahdi et al., 2022*). However, these studies did not evaluate the extent of readiness and perceptions of community pharmacists in Pakistan towards the telepharmacy services, particularly during the era of COVID-19 pandemic.

The majority of the pharmacists agreed that telepharmacy could improve medication adherence and disease therapy management among patients. The results of our study were similar to the earlier study conducted by *Hudd & Tataronis (2011)*, and another study conducted by *Pathak et al. (2020)*. These two studies did not find any difference in adherence rates between patients at a retail chain pharmacy and an urban telepharmacy-supported pharmacy. This means that telepharmacy can provide a quality service similar to physical visits to the community pharmacies and improve patients' quality of life (*Hudd & Tataronis, 2011*; *Pathak et al., 2020*). These results are also supported by a study conducted by the United States Veterans Affairs (VA) integrated health care system, which used clinical pharmacist-led synchronous clinical video telehealth (CVT). It was carried out for patients who needed anticoagulation services in a community outpatient clinic and found that the achievement of the international therapeutic normalized ratio (INR) remained stable between previous face-to-face management and CVT and this was reflected in higher levels of patient satisfaction (*Singh, Accursi & Black, 2015*).

In this study, we noted that pharmacists play an important role during the ongoing COVID-19 pandemic, particularly in Pakistan where healthcare resources are limited.

During the COVID-19 pandemic, most of the healthcare professionals reluctant to interact with patients amid fear of contracting the infection. Since the number of healthcare professionals in Pakistan is small, the implementation of telepharmacy and telehealth can be beneficial to overcome the issues faced in hospitals and pharmacy setups (*Alexander et al., 2017*). In addition, clinical pharmacists can continue to develop and implement strategies to improve treatment as a component of telehealth (*Badowski et al., 2018*). By implementing telepharmacy, the burden on healthcare professionals could be shared, which provides a positive impact on health care facilities. However, a concerted, team-based practice model in telehealth and telepharmacy is essential to provide coordinated and patient-centered care (*Poudel & Nissen, 2016*).

In the current study majority of the pharmacist believe that telepharmacy services can be provided to geriatric, bedridden patients and all those patients who use multiple medications for chronic illnesses. In literature, studies mentioned that geriatric, and bedridden patients faced transportation problems during their routine visit to doctors and pharmacist for medication counselling (*Le, Toscani & Colaizzi, 2020*). A previous study conducted in the United States of America (USA) launched an information system known as SinfoniaRx. This initiative was used to classify and determine possible drug-related needs of the patients. The team members asked the patients regarding their problems and address the concerns properly so their quality of life increased and medical cause decreased (*Bonner, 2016*).

Most of the respondents agreed that the telepharmacy services should only be provided by the clinical pharmacist along with good collaboration between physicians and other healthcare professionals. Pharmacists have an active role in providing pharmacy services to achieve the highest quality of care and for the public's safety, protection, and welfare in the use of pharmaceuticals (*Kimber & Peterson, 2006*; *Peterson & Anderson, 2004*). In several hospitals, the implementation of telepharmacy has led to a reduction in the rates of medication errors (*Casey et al., 2010*). Furthermore, selecting a pharmacist with strong clinical experience and interactive skills seems indispensable for what could be a particularly profound patient interaction (*Livet et al., 2021*; *Kerr et al., 2017*; *Sisler et al., 2019*).

Globally, the COVID-19 pandemic causes a tremendous disruption in all fields of normal life. During the delirious healthcare system, telepharmacy implementation could offer counseling services to the patients and the majority of the pharmacists agreed that telepharmacy implementation can help in providing optimal care to the patients. The world has officially extended the role of community pharmacists. Currently, pharmacists are involved in collaborating with customers and patients remotely using various telepharmacy tools (social media, virtual consultation, and home delivery services of medications) (*Mohamed Ibrahim et al., 2020*). The controlling of COVID-19 virus to cause infection is the utmost strategy in the current pandemic situation. Our respondents agreed that telepharmacy could positively impact the delivery of pharmaceutical care by limiting the transmission of diseases. Moreover, the enactment of telepharmacy can assist in escalating the pharmacist roles in the emergency response to include enhancing the public's awareness towards the symptoms of COVID-19 and direct them to a suitable

health care facility for testing and provide the essential information to decrease the virus spreading in the community.

## Limitations and strengths

This study is accompanied by a few limitations which should be considered while interpreting the results. Although this study achieved the estimated sample size but still, we believe that a large sample may provide slightly different results. Considering the government mandates towards COVID-19 and movement control measures, this study has less participation from the two provinces. However, the study population has representation from all major provinces and findings can be generalized to the other parts of the country. Since the survey link was shared through various social media applications, pharmacists who are not using these applications may not be represented by the study population. This sampling bias is aggressively addressed by sending survey links to the officials of pharmacy associations and societies which have WhatsApp groups of registered pharmacists at national and provincial levels. However, the selection bias cannot be disregarded in this study. Future studies must consider a simple or stratified random sampling to avoid any possible selection bias. Pharmacists may feel reluctant to answer the questions honestly amid the threats of professional self-esteem, reputation, and perceiving them as incompetent. We tried our best to mitigate this bias by administering the survey in an anonymous fashion. In addition, we requested the participants to avoid any information source to get the answer in order to ensure the assessment of their inherent perception and attitude towards the telepharmacy. Nevertheless, this study provides an important insight regarding the pharmacists' readiness and perception towards telepharmacy, which is currently a least appreciated area in Pakistan. In addition, this study is strengthened first study of its own kind including pharmacists across the country. The findings of the current study will aid health authorities to design and implement policies for telepharmacy in Pakistan.

## CONCLUSIONS

The majority of pharmacists had a positive perception regarding the eligibility of patients and regulatory issues/legal framework regarding the implementation of telepharmacy, as well as its use during the COVID-19 pandemic and beyond. Hence, it is proved that the implementation of telepharmacy can play a vital role in providing high-quality patient care with the additional advantage of being cost-effective. Whereas negative perception was seen regarding the benefits of telepharmacy implementation that must be addressed in future educational, awareness, and counseling programs. It is important for pharmacists to have latest authentic information about the benefits of telepharmacy implementation and to further convey this knowledge and belief to the community. In the era of shortage of healthcare providers in general and clinical pharmacists in particular, and the prevalence of COVID-19 today, telepharmacy can be an essential service to consider. Therefore, better-structured programs for pharmacists should be arranged to build an equilibrium in clinical knowledge about telepharmacy implementation. This study recommends that the health ministry and other associated authorities promote awareness of the

implementation of telepharmacy with a comprehensive training program. Moreover, telepharmacy services can be provided in any time of the day and delivered to remote patient centers or crowded environments which reduces the transmission of different infectious diseases and hence helps in the prevention/transmission of pandemics as well.

### Funding
This work was funded by Deanship of Scientific Research at Jouf University under Grant Number (DSR-2021-01-0333). The funders had no role in study design, data collection and analysis, decision to publish, or preparation of the manuscript.

### Grant Disclosures
The following grant information was disclosed by the authors:
Deanship of Scientific Research at Jouf University: DSR-2021-01-0333.

### Competing Interests
The authors declare that they have no competing interests.

### Author Contributions
- Khayal Muhammad conceived and designed the experiments, authored or reviewed drafts of the paper, and approved the final draft.
- Mohamed A. Baraka conceived and designed the experiments, analyzed the data, authored or reviewed drafts of the paper, and approved the final draft.
- Syed Sikandar Shah conceived and designed the experiments, analyzed the data, authored or reviewed drafts of the paper, and approved the final draft.
- Muhammad Hammad Butt conceived and designed the experiments, authored or reviewed drafts of the paper, and approved the final draft.
- Haytham Wali performed the experiments, authored or reviewed drafts of the paper, and approved the final draft.
- Muhammad Saqlain performed the experiments, analyzed the data, prepared figures and/or tables, and approved the final draft.
- Tauqeer Hussain Mallhi conceived and designed the experiments, authored or reviewed drafts of the paper, and approved the final draft.
- Khezar Hayat performed the experiments, analyzed the data, prepared figures and/or tables, and approved the final draft.
- Khairi Mustafa Fahelelbom analyzed the data, prepared figures and/or tables, and approved the final draft.
- Royes Joseph analyzed the data, prepared figures and/or tables, and approved the final draft.
- Yusra Habib Khan conceived and designed the experiments, authored or reviewed drafts of the paper, and approved the final draft.

## Ethics

The following information was supplied relating to ethical approvals (*i.e.*, approving body and any reference numbers):

The Riphah International University approved this study (RIP/271/0203).

## Data Availability

The raw measurements are available in the Supplemental File.

## Supplemental Information

Supplemental information for this article can be found online at http://dx.doi.org/10.7717/peerj.13296#supplemental-information.

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
