# Peer review of "Exploring the perception and readiness of Pharmacists towards telepharmacy implementation; a cross sectional analysis"

_PeerJ, doi:10.7717/peerj.13296_

## Round 0.1 · original submission · Major Revisions

Thanks for submitting your manuscript. Please make the necessary changes suggested by the reviewers. In addition to them, Line 227-228, the statement is not relevant. Generally, pharmacists are responsible for running a pharmacy, not doctors. Similarly, there are some contradicting statements in your manuscript. Could you please recheck it?

Secondly, present the measures separately in the methodology with clear explanation.

Reviewer 1 ·

Basic reporting

The introduction part of the study is well explained. but I would like to suggest you to improve the description in the very first paragraph, in order to avoid the content repetition.

Experimental design

even though it had some limitations on the no. of participants, the methodological part can be well accepted.

Validity of the findings

your study primarily aimed to assess the perception of pharmacist towards telecommunications and the findings that obtained through the statistical analysis are similiar to the same type of study conducted in other countries as well. statistical analysis can be well accepted.

Additional comments

As a reader i would like to know that whether tele pharmacy is adapted in any part of the country, as this is as an innovative study in Pakistan for assessing the perception of pharmacist towarrds implementation of telepharmacy.

Annotated reviews are not available for download in order to protect the identity of reviewers who chose to remain anonymous.

Reviewer 2 ·

Basic reporting

Thu authors conducted a study on exploring the perception and readiness of Pakistani
pharmacists towards telepharmacy implementation. This is a timely research. I highly appreciated the authors effort to conduct such an important study. This is a good piece of work. The manuscript has a good potential and can be accepted after further revision. I have some observation which I think could help the author to improve the manuscript. My ocomments are below:

1. The authors have done interesting work. Congratulation on your work on this topical issue.

2. The abstract needs to be improved. I would suggest the authors follow the order of: What is the problem? Why is the problem important? What have others done to solve the problem? What are you doing to solve the problem? In addition, expand a bit of the method part in the abstract. Please remove the software name from the abstract section.


3. The authors provided a narrative description of telepharmacy and its importance in healthcare service, especially for pharmacists. They also add a short literature review from lines 92-101. However, it was difficult to get a depth idea about the topics with such a short literature review. Include a literature review section where you present the results of earlier studies from the field.

4. The English language should be improved to ensure that an international audience can clearly understand your text. In particular, the Abstract, Introduction, Method, and Result section need major revision of English language editing. I suggest you have a colleague who is proficient in English and familiar with the subject matter review your manuscript.

Experimental design

5. The article is original and within the scope of this journal.

6. The author nicely articulated the evolution of telemedicine and its importance in healthcare service worldwide in the introduction section. They also explained telepharmacy which is the main topic of this current study. However, the authors did not indicate in the introduction section if this was the first study in Pakistan. Then, what was the novelty of this study? The author explained very little about the justification of the study which is not sufficient to understand the motivation to conduct this study. Need a clear explanation of the knowledge gap and addressing approach of this study. Therefore, I recommend adding a separate paragraph on the justification of the study by mentioning the knowledge gap and how to address them in this study. Please also explicitly mention the hypothesis or research question/objective if any. I suggest authors organize the introduction section from a more general to a more specific character.

7. The ethical standard is ok.

8. The study designs and methodology are well explained. In the method section, the authors should clearly specify the sampling technique used in this study so that it is clear to the reader. For the online study, many authors used convenience sampling while others use the snowball sampling technique. However, both are associated with section bias. Thus, can you please explain how did you avoid the selection bias in your study?

9. In method, the study instrument (2nd, 3rd, 4th, and 5th sections) used in this study was globally validated? If yes, please use a valid reference (otherwise it does not make sense). If not, how did you validate the survey items? If the survey item was author-produced, please estimate the internal consistency of the items.

10. Why the authors did not consider other sociodemographic factors such as urbanicity, education, monthly income, area of work, job experience as a pharmacist which could have a major influence on the telepharmacy implementation?

11. Please expand the statistical analysis section a bit (perhaps you can add why you used the chi-square test).

Validity of the findings

12. In the results section, the data presented in the tables are easily understandable and represent the tests performed. Highly Appreciated.

13. In Table 1, to present “age”, you mentioned both percentages and mean (sd). Please use only one of them.

14. In the discussion section, the author provided a narrative interpretation of the results but lack an explicit discussion on comparing present study findings with previous studies. Therefore, I suggest to expand the discussion accordingly (Answer your objective/hypotheses and link your results to the results of earlier studies in the field).

15. Add a separate section on implication and policy recommendation.

16. Add some strengths to your study.

17. The limitation section must be improved. All limitations such as sampling biases, response bias, cross-sectional study consequences, other factors consideration, etc should be mentioned. Also, write about the future line of research on this topic.

Additional comments

No further comments.

---

## Round 0.2 · accepted · Accept

Thanks for making the changes.

Reviewer 1 ·

Basic reporting

no comments

Experimental design

no comments

Validity of the findings

no comments

Reviewer 2 ·

Basic reporting

The authors have substantially improved this section.

Experimental design

The authors have substantially improved this section.

Validity of the findings

The findings have well-validated.